# Preoperative CT anthropometric measurements and pancreatic pathology increase risk for postoperative pancreatic fistula in patients following pancreaticoduodenectomy

**Yun Hwa Roh**[1], **Bo Kyeong Kang**[1], **Soon-Young Song**[1], **Chul-Min Lee**[1], **Yun Kyung Jung**[2], **Mimi Kim**[1] *

**1** Department of Radiology, Hanyang University College of Medicine, Seoul, Republic of Korea,
**2** Department of Surgery, Hanyang University College of Medicine, Seoul, Republic of Korea

* bluefish010@naver.com

## Abstract

Postoperative pancreatic fistula (POPF) is a common complication following pancreatico-duodenectomy (PD). However, risk factors for this complication remain controversial. We conducted a retrospective analysis of 107 patients who underwent PD. POPF was diagnosed in strict accordance with the definition of the 2016 update of pancreatic fistula from the International Study Group on Pancreatic Fistula (ISGPF). Univariate and multivariate logistic regression analyses were performed to identify independent risk factors for POPF. A total of 19 (17.8%) subjects of pancreatic fistula occurred after PD, including 15 (14.1%) with grade B POPF and 4 (3.7%) with grade C POPF. There were 33 (30.8%) patients with biochemical leak. Risk factors for POPF (grade B and C) were larger area of visceral fat (odds ratio [OR], 1.40; $p$ = 0.040) and pathology other than pancreatic adenocarcinoma or pancreatitis (OR, 12.45; $p$ = 0.017) in the multivariate regression analysis. This result could assist the surgeon to identify patients at a high risk of developing POPF.

## Introduction

Pancreaticoduodenectomy (PD) is a standard procedure for patients with benign or malignant tumors involving the head of the pancreas and periampullary regions [1–3]. Attributable to advancements in surgical techniques and perioperative management, the mortality rate after PD has improved significantly, reportedly reaching 1–2% at high volume centers [4–6]. However, the postoperative morbidity rate remains high, ranging from 27.1% to 43% [6–8].

Among the reported complications, postoperative pancreatic fistula (POPF) is the most common complication following PD and is associated with delayed gastric emptying, intra-abdominal abscess and hemorrhage, and superimposed infection and sepsis, consequently increasing the length of stay and even leading to reoperation in some cases [9–11]. Therefore, reducing the rate of POPF after PD is a serious challenge for clinicians.

**Data Availability Statement:** All relevant data are within the paper and its Supporting Information files.

**Funding:** This work was supported by the research fund of Hanyang University (HY-2019). The funders had no role in study design, data collection and analysis, decision to publish, or preparation of the manuscript.

**Competing interests:** The authors have declared that no competing interests exist.

Various risk factors for POPF have been suggested, including sex, body mass index (BMI), pancreatic duct size, pancreatic texture, blood transfusion, intraoperative blood loss, operation time, and visceral fat area [9,12,13]. However, definite causal factors for developing POPF after PD remain controversial. Recently, some studies have suggested a relationship between surgical outcomes and anthropometric measurements, such as the core muscle mass and body fat area. Depleted skeletal muscle mass and visceral obesity increased postoperative complications after total gastrectomy, major hepatectomy, and colorectal cancer [14–16]. With respect to PD, patients with sarcopenia and visceral obesity showed decreased survival and increased morbidity [12,17,18]. As the POPF criteria were updated in 2016, biochemical leak (POPF A) is no longer considered true POPF [19]. A few studies had reported on the association of anthropometric measurement with POPF according to the revised criteria [20,21].

In this study, we performed a retrospective study to evaluate the association and predictive value of anthropometric measurements and other pre- and perioperative variables for POPF.

## Methods

### Subjects

This retrospective study was approved by the institutional review board (IRB) of Hanyang University Hospital. All experiments were performed in accordance with the relevant guidelines and regulations.

One-hundred and twenty-four consecutive subjects who underwent PD between October 2007 and October 2017 were enrolled into this study. The exclusion criteria were as follows: lack of clinical information (hardness of pancreas [n = 9], BMI [n = 1]); interval between preoperative computed tomography (CT) and surgery >40 days (n = 3); unavailability of preoperative CT (n = 3); and difficulty evaluating POPF (n = 1). Finally, 107 subjects (male: female = 64:43; mean age, 65.9 years; range, 35–82 years) who underwent PD were included. Detailed information was obtained from electronic medical records. For each subject, the following data were collected: (1) subjects' demographic and clinical features, including age, sex, BMI, preoperative albumin, and total bilirubin; (2) operative details, including pancreatic texture, operation time, performance of intraoperative blood transfusion, and the use of a pancreatic stent; and (3) final diagnosis of the tumor. The pancreatic texture of all subjects was examined by the surgeon during the operation and classified as either soft or hard. A pancreatic duct stent was occasionally used during reconstruction following PD according to the surgeon's judgment. External drains were inserted in proximity to the pancreatic anastomosis for each surgery, and the amylase level of drain fluid was routinely measured during the inserted period.

### Measurement of anthropometric measurements

We measured subjects' abdominal circumference, visceral and subcutaneous fat, and total abdominal muscle area on preoperative CT scans at the level of the third lumbar vertebra (L3). Distinction among the muscle, fat, and different tissues was based on Hounsfield units (HU) using AquariusNET Server (TaraRecon, Foster City, CA, USA). A threshold range of -29 to 150 HU was used to define muscle and a range of -190 to -30 HU was used to define fat. Hand adjustment of the selected area was performed (Figs 1 and 2). Skeletal muscle mass was normalized for the subjects' heights to calculate the skeletal muscle mass index (SMI, $cm^2/m^2$). The ratio of visceral fat to SMI (VF/SMI) was also calculated. Sarcopenia was defined as SMI $\leq 52.4$ $cm^2/m^2$ for men and $\leq 38.5$ $cm^2/m^2$ for women based on a study by Prado et al. These cutoff values are accepted by an international consensus group on the diagnostic criteria for cachexia associated with cancer [22,23]. Visceral obesity was defined as a visceral fat area $\geq 100$

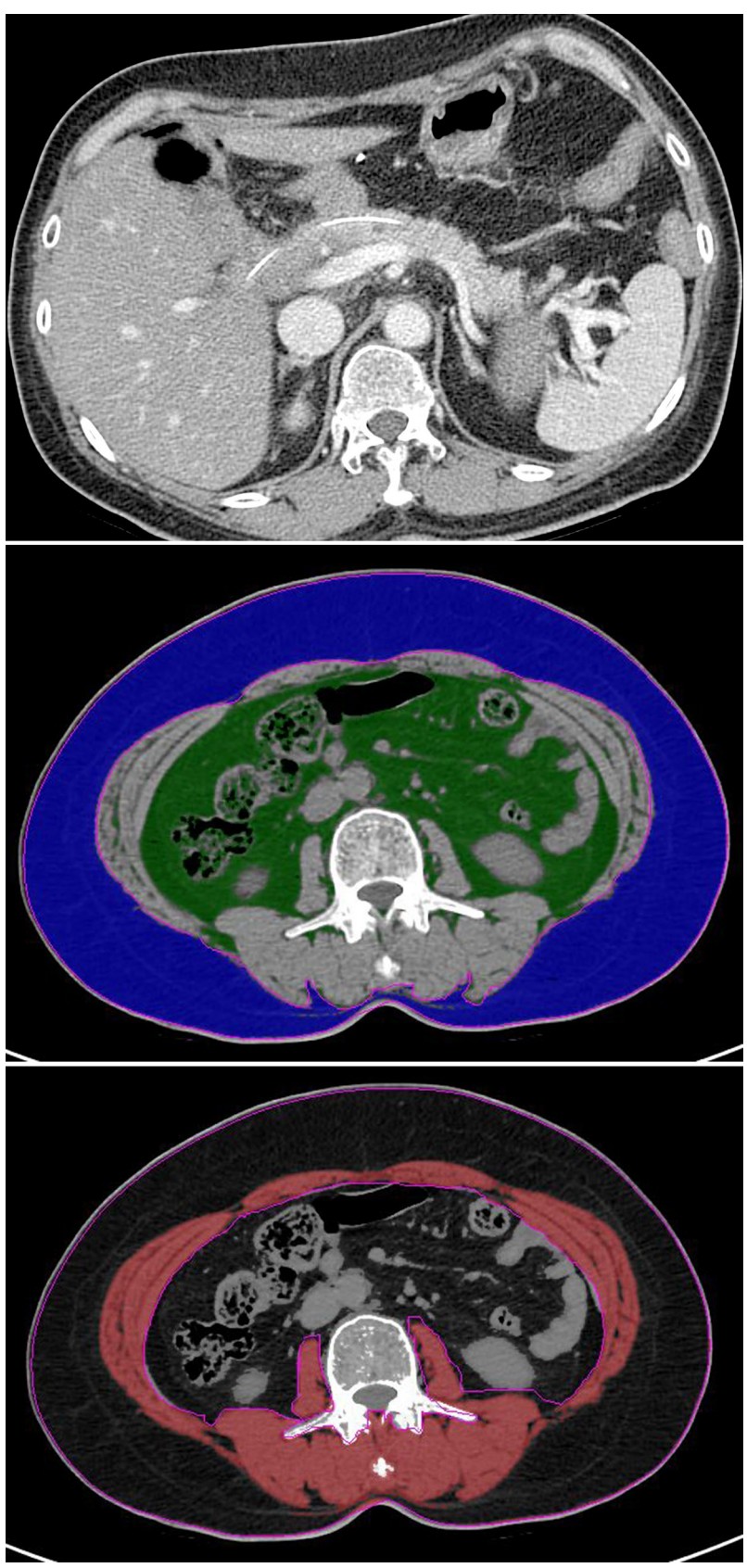

**Fig 1. A 58-year-old woman treated with pancreaticoduodenectomy for ampulla of Vater cancer without POPF.**
(A) Axial contrast enhanced CT taken 8 days following surgery show small amount of fluid collection around pancreaticojejunal anastomosis, but the amylase level in the drained fluid is not greater than three times the upper normal serum value. (B,C) On the preoperative axial CT, the scan was segmented into subcutaneous fat in blue, total abdominal muscle area in read and visceral fat area in green. The patient shows a visceral fat area of 76 cm$^2$ and VF/SMI of 1.6.

cm$^2$ in both sexes. This value is widely used as a cutoff to define sarcopenic obesity in Asian populations and is equivalent to that used for the diagnosis of metabolic syndrome in Japan [24,25].

## Definition of postoperative pancreatic fistula

Pancreatic fistula was defined according to the revised 2016 International Study Group on Pancreatic Fistula (ISGPF) classification and grading [19]. The previous 'grade A' POPF is newly classified as 'biochemical leak', which refers to a transient and asymptomatic biochemical fistula. Grade B and C POPF are clinically relevant fistulae, which require a change in postoperative management. If peripancreatic drainage persists for more than 3 weeks or is repositioned through interventional procedures, it is classified as Grade B. Grade C POPF refers to those with POPF-related organ failure, reoperation, or death.

## Statistical analysis

Normally distributed numerical variables are presented as mean and standard deviation (SD) and were compared using the independent t-test, and non-normally distributed numerical variables are presented as median (the first quartile–the third quartile) and were compared using the Mann-Whitney test. Categorical variables are presented as frequencies with percentage and were tested using the chi-squared test or Fisher's exact test. Univariate logistic analysis and backward stepwise multivariate logistic regression analysis were performed to identify the independent risk factors for POPF after PD. Variables with $P$-values of <0.05 on univariate analyses were entered into the final multivariate model to reveal risk factors for POPF. For each parameter, an odds ratio (OR) for POPF was provided with a 95% confidence interval (CI). All statistical analyses were performed using SAS version 9.4 (SAS Institute, Cary, NC). $P$-values of <0.05 were considered statistically significant.

## Results

Demographic and clinical characteristics are shown in Table 1. The median time interval between preoperative CT and surgery was 14 days (mean, 15 days; range, 3–40 days). A total of 19 (17.8%) patients developed a pancreatic fistula after PD, including 15 (14.1%) with grade B POPF, and 4 (3.7%) with grade C POPF. In this study, all patients with POPF C underwent reoperation. There were 33 (30.8%) patients with biochemical leak.

The area of visceral fat, VF/SMI ratio, pancreas hardness and pathology were significantly different between the two groups (Table 2). Subjects with POPF had a significantly larger area of visceral fat (159.6 cm$^2$ vs. 120.3 cm$^2$, $p$ = 0.022), higher VF/SMI ratio (3.30 vs. 2.54, $p$ = 0.030), more frequently had a soft pancreas (78.9% vs. 53.4%, $p$ = 0.045), and more frequently had pathology other than pancreatic adenocarcinoma or pancreatitis (97.4% vs. 62.5%, $p$ = 0.045) than patients without POPF. However, sex, BMI, pancreatic duct diameter, operation time, intraoperative blood transfusion and use of a pancreatic stent were not significantly different between the two groups.

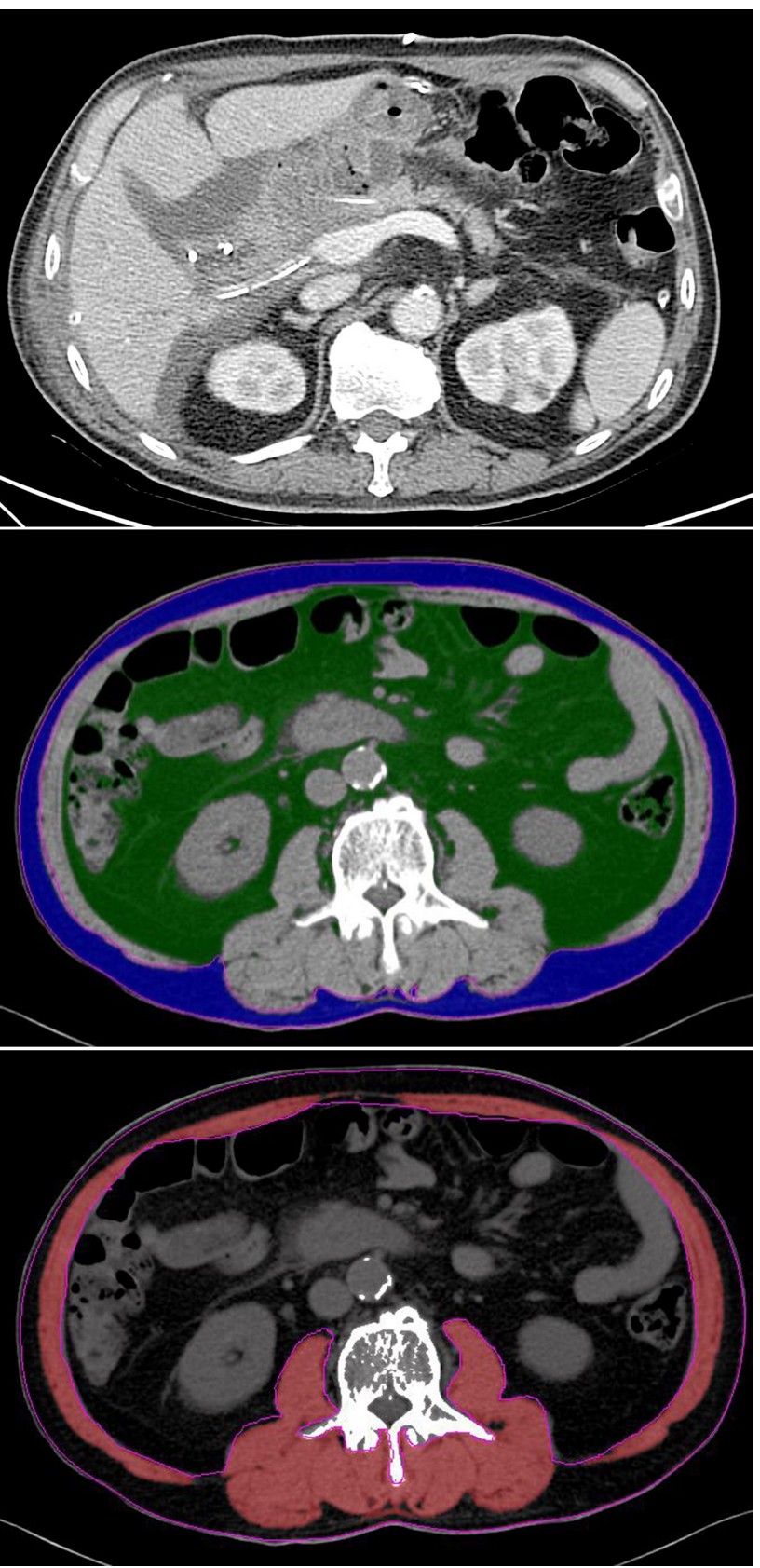

**Fig 2. A 65-year-old man treated with pancreaticoduodenectomy for ampulla of Vater cancer.** (A) Axial contrast enhanced CT taken 7 days after surgery show fluid collection around the pancreaticojejunal anastomosis with suspicious dehiscence. The patient was treated with percutaneous drainage of fluid collection around the anastomosis. (B,C) On the preoperative axial CT, the patient shows a visceral fat area of 240 cm$^2$ and VF/SMI of 5.8.

Univariate regression analysis showed significant correlation between POPF and the following factors (Table 3): higher visceral fat (odds ratio [OR]: 1; 95% confidence interval [CI]: 1.00–1.02, $p = 0.026$), higher VF/SMI ratio (OR: 1.46; 95% CI: 1.03–2.07; $p = 0.036$), a soft texture of the pancreas (OR: 3.27; 95% CI: 1.01–10.64; $p = 0.049$), and pathology other than pancreatic adenocarcinoma or pancreatitis (OR: 2.38; 95% CI: 1.38–84.69; $p = 0.024$).

In the multivariate regression analysis, higher visceral fat (OR: 1.40; 95% CI: 1.07–15.43, $p = 0.040$) and pathology other than pancreatic adenocarcinoma or pancreatitis (OR: 12.45; 95% CI: 1.59–99.28; $p = 0.017$) were identified as independent risk factors for POPF (Table 4).

## Discussion

Pancreatic fistula after PD remains a challenging problem, even in high-volume centers. Identification of patients at a high risk of developing pancreatic fistula helps in a more elaborate risk-benefit assessment before surgery and may allow clinicians to coordinate perioperative care. In our study, we observed POPF in 17.8% (19/107) of patients. Higher visceral fat and pathology other than pancreatic adenocarcinoma or pancreatitis were independent risk factors for developing POPF after PD.

Recently, studies regarding the effects of sarcopenia and visceral obesity on POPF have been conducted. To date, few studies have examined the impact of sarcopenic obesity on

**Table 1. Baseline characteristics of subjects.**

| Characteristics | |
|---|---|
| Demographic data | |
| Subjects, no. | 107 |
| Age, years | 65.9 ± 9.9 |
| Male, no. (%) | 64 (59.8) |
| Body mass index, kg/m$^2$ | 23.2 ± 3.0 |
| Final diagnosis, no. (%) | |
| Benign disease | 10 (9.3) |
| Chronic pancreatitis | 3 |
| Pseudocyst | 1 |
| Malignant disease | 97 (90.7) |
| CBD cancer | 34 |
| Pancreatic head cancer | 31 |
| AOV cancer | 23 |
| Intraductal papillary mucinous neoplasm | 7 |
| Duodenal cancer | 2 |
| POPF, no. (%) | |
| Absence | 55 (51.4) |
| Biochemical leak | 33 (30.8) |
| POPF grade B | 15 (14.1) |
| POPF grade C | 4 (3.7) |

Note. Data are presented as mean ± standard deviation or number of subjects with percentage in parentheses.

**Table 2. Subject demographics and clinical characteristics.**

| | POPF (-) (n = 88) | POPF (+) (n = 19) | *p*-value |
|---|---|---|---|
| Age, years | 65.8 ± 9.9 | 65.9 ± 10.4 | 0.983 |
| Sex | | | 0.074 |
| Male, no. (%) | 49 (55.7) | 15 (78.9) | |
| Female, no. (%) | 39 (44.3) | 4 (21.1) | |
| Body mass index, kg/m$^2$ | 23.1 ± 2.9 | 23.9 ± 3.2 | 0.273 |
| Preoperative albumin, g/dL | 3.75 ± 0.52 | 3.9 ± 0.59 | 0.085 |
| Preoperative total bilirubin, mg/dL (IQR) | 1.73 (0.69–4.28) | 1.90 (0.58–5.80) | 0.453 |
| Pancreatic duct diameter, mm (IQR) | 3.5 (1–6) | 2 (1–4) | 0.858 |
| Skeletal muscle index (SMI), cm$^2$ | 46.9 ± 9.1 | 47.2 ± 9.8 | 0.902 |
| Visceral fat (VF), cm$^2$ | 120.3 ± 62.2 | 159.6 ± 84.8 | 0.022 * |
| Subcutaneous fat, cm$^2$ | 109.7 ± 59.3 | 104.3 ± 57.3 | 0.718 |
| Abdominal circumference, cm | 83.3 ± 8.5 | 87.3 ± 9.5 | 0.067 |
| VF/SMI | 2.54 ± 1.32 | 3.30 ± 1.57 | 0.030 * |
| Sarcopenia, no. (%) | | | 0.493 |
| No | 40 (45.5) | 7 (36.8) | |
| Yes | 48 (54.5) | 12 (63.2) | |
| Visceral obesity, no. (%) | | | 0.067 |
| No | 34 (38.6) | 3 (15.8) | |
| Yes | 54 (61.4) | 16 (84.2) | |
| Pancreatic hardness, no. (%) | | | 0.045 * |
| Soft | 47 (53.4) | 15 (78.9) | |
| Hard | 41 (46.6) | 4 (21.1) | |
| Operation time, minutes | 441 ± 71 | 476 ± 71 | 0.051 |
| Transfusion, no. (%) | | | 0.481 |
| No | 54 (61.4) | 10 (52.6) | |
| Yes | 34 (38.6) | 9 (47.4) | |
| Stent, no. (%) | | | 0.207 |
| No | 11 (12.5) | 0 | |
| Yes | 77 (87.5) | 19 (100) | |
| Pathology, no. (%) | | | 0.006* |
| PDAC or pancreatitis | 33 (37.5) | 1 (5.3) | |
| Other pathology | 55 (62.5) | 18 (94.7) | |

Note. Data are presented as mean ± standard deviation, median with interquartile range, or number of subjects with percentage in parentheses. * are the parameters with p<0.05.

survival in patients with pancreaticobiliary tumors [20,21,26,27]. Obesity and sarcopenia were synergistic and believed to exacerbate the risk of death, as well as the risk of metabolic disorders, and the number of related studies is increasing [28].

In our study, a larger area of visceral fat was significantly associated with POPF in both univariate and multivariate analyses, consistent with previous studies. The frequency of visceral obesity was higher in patients with POPF (84.2%, 16/19) compared to those without POPF (61.4%, 54/88), although it was not statistically significant. Percorali et al. also reported that the visceral fat area was an independent predictor of pancreatic fistula in patients undergoing PD [12]. Generally, patients with greater subcutaneous and visceral fat accumulation provide greater technical difficulty for surgeons. The view of the surgical field is deeper and poorer in obese patients, which may increase the risk of pancreatic fistula [14,29,30]. Other than

**Table 3. Univariate risk factor analysis for postoperative pancreatic fistula.**

| Variable | Odds ratio | 95% C.I. | p-value |
|---|---|---|---|
| Sex [Female] | 2.99 | 0.92, 9.72 | 0.069 |
| BMI | 1.1 | 0.93, 1.31 | 0.272 |
| Preoperative albumin | 2.23 | 0.88, 5.63 | 0.9 |
| Preoperative total bilirubin | 1.06 | 0.94, 1.20 | 0.326 |
| Pancreatic duct diameter | 0.99 | 0.86, 1.13 | 0.856 |
| SMI | 1 | 0.95, 1.06 | 0.901 |
| Visceral fat | 1 | 1.00, 1.02 | 0.026 * |
| Subcutaneous fat | 0.99 | 0.99, 1.01 | 0.715 |
| Abdominal circumference | 1.56 | 0.99, 1.12 | 0.07 |
| VF/SMI | 1.46 | 1.03, 2.07 | 0.036 * |
| Sarcopenia [Absence] | 0.7 | 0.25, 1.95 | 0.494 |
| Visceral obesity [Absence] | 0.3 | 0.08, 1.10 | 0.069 |
| Pancreatic hardness [Hard] | 3.27 | 1.01, 10.64 | 0.049 * |
| Operation time | 1.01 | 1.00, 1.01 | 0.055 |
| Transfusion [Absence] | 0.7 | 0.26, 1.90 | 0.483 |
| Stent [Absence] | 0 | 0 | 0.999 |
| Pathology [PDAC or pancreatitis] | 2.38 | 1.38, 84.69 | 0.024* |

Note. Reference categories are in square brackets.

* are the parameters with p<0.05. C.I.: Confidence interval, BMI: Body mass index, PD: Pancreatic duct, VF: Visceral fat, SMI: Skeletal muscle index.

mechanical reasons, excessive visceral fat is associated with insulin resistance [31] and comorbidities such as type 2 diabetes, atherosclerosis and cardiovascular disease [32,33], which may affect surgical outcomes negatively, including a higher rate of wound infection and anastomotic fistula and a longer hospital stay [30,34,35].

The skeletal muscle index and sarcopenia showed little association with POPF in this study. In a meta-analysis by Ratnayake et al., preoperative sarcopenia was not a significant negative predictive factor in postoperative morbidity, including POPF, following pancreatic resection [21]. On the contrary, there are several studies showing that sarcopenia is an independent risk factor for POPF [36,37]. Meanwhile, VF/SMI ratio was higher in POPF group than non-POPF group in our study, although it was not significant at multivariate analysis. High VF/SMI ratio could be referred to as sarcopenic obesity, a condition in which loss of muscle mass is accompanied by increase in fat accumulation [22]. Jang et al. also examined 284 patients who underwent PD between 2005 and 2016, concluded that sarcopenic obesity was the only predictor for POPF [20]. Although our study and study by Jang et al. used predefined cutoff for visceral obesity, sarcopenia, and sarcopenic obesity, an accurate definition of sarcopenic obesity has not

**Table 4. Multivariable risk factor analysis for postoperative pancreatic fistula.**

| Variable | Odds ratio | 95% C.I. | p-value |
|---|---|---|---|
| Visceral fat | 1.40 | 1.07, 15.43 | 0.040* |
| Pathology [PDAC or pancreatitis] | 12.45 | 1.59, 99.28 | 0.017* |

Note. Reference categories are in square brackets. C.I.: Confidence interval, VF/SMI: Visceral fat to skeletal muscle index ratio.

* are the parameters with p<0.05.

yet been established. These various conclusions from existing studies suggest that the impact of sarcopenia and sarcopenic obesity in developing POPF following PD is still inconclusive and therefore further research is needed.

Higher POPF rate was observed in pathology other than PDAC or chronic pancreatitis, which was categorized according to the previously established fistula risk score criteria [38]. The mechanism by which pathologies other than PDAC or chronic pancreatitis increases the risk of POPF is related to the effect of soft pancreas parenchyma. Pathologies with pancreatic adenocarcinoma and chronic pancreatitis are more likely to result in hard parenchyma, therefore there could be a benefit in reducing the incidence of POPF. Hu et al. retrospectively analyzed 536 cases and also found that a soft pancreas was an independent risk factor for developing pancreatic fistula, including biochemical leak and POPF B and C, after PD (OR: 3.05, $p<0.001$) [9]. Kawai et al. reviewed 1,239 patients from 11 Japanese medical centers and concluded that a soft pancreas was a significant predictive factor for pancreatic fistula (OR: 2.7, $p = 0.001$) [39]. In this study, among a total of 107 patients, a soft pancreas was more frequent in subjects with POPF than in those without POPF (78.9% vs. 53.4%), although the result was not significant at multivariable analysis. A soft pancreas is more prone to laceration when performing suturing and tying. Moreover, the soft texture of pancreatic remnants induces technical difficulties in performing pancreatoenteric and duct-to-mucosa anastomoses, which increase the risk of anastomotic leakage [40]. Although it is not possible to evaluate soft pancreas before surgery, the possibility of POPF can be assessed by guessing the pathology through preoperative CT.

Our study had several limitations. First, due to its retrospective nature, it may have been influenced by selection and information biases. Second, it is a single-center study, which only includes Koreans; therefore, the findings may not be applicable to other populations. Third, several surgical anastomotic techniques have been established in recent years, but the technique used in each surgery was not analyzed in this study. Also, the pancreatic texture during the operation was evaluated by the surgeon, which may be subjective. Studies related to quantitative measurement of pancreas texture have been reported, such as MR elastography or ultrasound elastography [41,42]. Further research regarding the objective assessment of the pancreas texture is warranted.

In conclusion, larger area of visceral fat and pathology other than pancreatic adenocarcinoma or pancreatitis are independent predictors of POPF after pancreaticoduodenectomy and may allow better preparation of the postoperative care for patients by identifying patients at a high risk of developing pancreatic fistulas.

## Supporting information

**S1 Dataset.**
(XLSX)

**S1 Table. Comparison of clinical characteristics in subjects with and without visceral obesity.**
(DOCX)

## Author Contributions

**Conceptualization:** Mimi Kim.

**Data curation:** Yun Hwa Roh, Yun Kyung Jung.

**Formal analysis:** Bo Kyeong Kang, Mimi Kim.

**Methodology:** Soon-Young Song, Chul-Min Lee.

**Supervision:** Mimi Kim.

**Writing – original draft:** Yun Hwa Roh.

**Writing – review & editing:** Mimi Kim.

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
