## [Decision Letter · Decision Letter 0]

18 Aug 2020

PONE-D-20-24280

Predictive value of preoperative CT for postoperative pancreatic fistula in patients following pancreaticoduodenectomy

PLOS ONE

Dear Dr. Kim,

Thank you for submitting your manuscript to PLOS ONE. After careful consideration, we feel that it has merit but does not fully meet PLOS ONE’s publication criteria as it currently stands. Therefore, we invite you to submit a revised version of the manuscript that addresses the points raised during the review process.

We look forward to receiving your revised manuscript.

Kind regards,

Ulrich Wellner, PD Dr. med.

Academic Editor

PLOS ONE

Journal Requirements:

2. Please amend the manuscript submission data (via Edit Submission) to include author Soon-Young Song.

Reviewers' comments:

Reviewer's Responses to Questions

**Comments to the Author**

1. Is the manuscript technically sound, and do the data support the conclusions?

Reviewer #1: Yes

Reviewer #2: Partly

2. Has the statistical analysis been performed appropriately and rigorously? 

Reviewer #1: Yes

Reviewer #2: Yes

3. Have the authors made all data underlying the findings in their manuscript fully available?

Reviewer #1: Yes

Reviewer #2: Yes

4. Is the manuscript presented in an intelligible fashion and written in standard English?

Reviewer #1: Yes

Reviewer #2: Yes

5. Review Comments to the Author

Reviewer #1: In this article, the authors conducted a retrospective analysis of 107 patients who underwent pancreaticoduodenectomy (PD). They performed univariate and multivariate logistic regression analyses to identify independent risk factors for POPF. They demonstrated that risk factors for POPF (grade B and C) were larger area of visceral fat and pathology other than pancreatic adenocarcinoma or pancreatitis. This research provides us new insights of preoperative CT for postoperative pancreatic fistula in patients following pancreaticoduodenectomy. However, the study has some weakness and concerns before the paper can be published.

Major revisions:

1. For the “Material and methods” part, the authors did not explain the authors did not explain the uniform standard for tissue sample texture. Moreover, the pancreatic texture of all subjects was examined by the surgeon, which is too subjective.

2. In Table 4, those variable whose p-value was over 0.05 should be listed.

Reviewer #2: This is a study on the prediction and evaluation of pancreatic leakage after pancreatoduodenectomy, which has certain clinical application.However, in this study, the interval between CT detection and surgery was the longest about 50 days, which significantly affected the evaluation effect.If the interval is about a week, the credibility is much higher.

6. PLOS authors have the option to publish the peer review history of their article (what does this mean?). If published, this will include your full peer review and any attached files.

Reviewer #1: **Yes: **Weilin Wang

Reviewer #2: No

---

## [Author Response · Author response to Decision Letter 0]

10 Sep 2020

Reviewer #1.

1.For the “Material and methods” part, the authors did not explain the uniform standard for tissue sample texture. Moreover, the pancreatic texture of all subjects was examined by the surgeon, which is too subjective.

Recently, studies were conducted to objectify the pancreas texture or fatty pancreas through MR elastography or pancreas CT density measurement [1,2]. However, the analysis is complex and only possible in hospitals with special equipment. 

Although it is a retrospective study and the evaluation of pancreas texture was subjective, it could be meaningful. Because, in our hospital, a structured questionnaire was recorded during or immediately after surgery about the surgery by specialized pancreas surgeon as a routine process, and pancreas texture also recorded in the same process. In addition, although the widely accepted definition of pancreas texture is still insufficient, the evaluation of the pancreatic texture in two stage by subjective judgment is still acceptable in systematic review and meta-analysis [3].

2.In Table 4, those variable whose p-value was over 0.05 should be listed.

Thank you for your kind comments. The odds ratio and p-value of the remaining variables are not shown since the stepwise method for multivariable logistic regression analysis was used in the analysis. This is added to the statistical analysis in the text. 

Reviewer #2

3.This is a study on the prediction and evaluation of pancreatic leakage after pancreatoduodenectomy, which has certain clinical application. However, in this study, the interval between CT detection and surgery was longest about 50 days, which significantly affected the evaluation effect. If the interval is about a week, the credibility is much higher.

Thank you for your kind comment. Previous study of sarcopenia showed that CT scan taken within 30 days were used for analysis [4]. In our study, the median time interval between preoperative CT and surgery was 14 days (mean, 15 days; range, 3-40 days). Although if the Interval was shorter, it would be expected to have higher credibility, it could not be done due to limited subjects. The interval between preoperative CT and surgery was added in the result part. 

References

1. Shi Y, Liu Y, Gao F, Liu Y, Tao S, Li Y et al. Pancreatic Stiffness Quantified with MR Elastography: Relationship to Postoperative Pancreatic Fistula after Pancreaticoenteric Anastomosis. Radiology. 2018 Aug;288(2):476-484. 

https://doi.org/10.1148/radiol.2018170450 PMID: 29664337 

2. Fukuda Y, Yamada D, Eguchi H, Hata T, Iwagami Y, Noda T et al. CT Density in the Pancreas is a Promising Imaging Predictor for Pancreatic Ductal Adenocarcinoma. Ann Surg Oncol. 2017;24(9):2762-2769. https://doi.org/10.1245/s10434-017-5914-3 PMID: 28634666

3. Eshmuminov D, Schneider MA, Tschuor C, Raptis DA, Kambakamba P, Muller X et al. Systematic review and meta-analysis of postoperative pancreatic fistula rates using the updated 2016 International Study Group Pancreatic Fistula definition in patients undergoing pancreatic resection with soft and hard pancreatic texture. HPB (Oxford). 2018;20(11):992-1003. https://doi.org/10.1016/j.hpb.2018.04.003 PMID: 29807807

4. Prado CM, Lieffers JR, McCargar LJ, Reiman T, Sawyer MB, Martin L, Baracos VE. Prevalence and clinical implications of sarcopenic obesity in patients with solid tumours of the respiratory and gastrointestinal tracts: a population-based study. Lancet Oncol. 2008;9(7):629-635. https://doi.org/10.1016/S1470-2045(08)70153-0 PMID: 18539529

---

## [Decision Letter · Decision Letter 1]

13 Oct 2020

PONE-D-20-24280R1

Predictive value of preoperative CT for postoperative pancreatic fistula in patients following pancreaticoduodenectomy

PLOS ONE

Dear Dr. Kim,

Thank you for submitting your manuscript to PLOS ONE. After careful consideration, we feel that it has merit but does not fully meet PLOS ONE’s publication criteria as it currently stands. Therefore, we invite you to submit a revised version of the manuscript that addresses the points raised during the review process.

We look forward to receiving your revised manuscript.

Kind regards,

Ulrich Wellner, PD Dr. med.

Academic Editor

PLOS ONE

Reviewers' comments:

Reviewer's Responses to Questions

**Comments to the Author**

1. If the authors have adequately addressed your comments raised in a previous round of review and you feel that this manuscript is now acceptable for publication, you may indicate that here to bypass the “Comments to the Author” section, enter your conflict of interest statement in the “Confidential to Editor” section, and submit your "Accept" recommendation.

Reviewer #3: All comments have been addressed

Reviewer #4: All comments have been addressed

2. Is the manuscript technically sound, and do the data support the conclusions?

Reviewer #3: Partly

Reviewer #4: Yes

3. Has the statistical analysis been performed appropriately and rigorously? 

Reviewer #3: Yes

Reviewer #4: Yes

4. Have the authors made all data underlying the findings in their manuscript fully available?

Reviewer #3: Yes

Reviewer #4: Yes

5. Is the manuscript presented in an intelligible fashion and written in standard English?

Reviewer #3: Yes

Reviewer #4: Yes

6. Review Comments to the Author

Reviewer #3: This is a well written paper looking for associations that lead to post operative pancreatic fistulas, which is an important sequelae following pancreaticoduodenectomy.

I have a few comments:

1. I agree with reviewer 1, that points out that pancreatic texture hardness/softness is a subjective measure, though from clinical experience I agree that a 'soft' pancreas is more likely to lead to a leak. I accept that the authors have responded to the reviewer's comments: I would add that perhaps the authors add a sentence or two in the limitations of their work that this is a subjective measure, and more work needs to be done to be able to characterise pancreatic tissue density (and also reference the work relating to MR elastogram etc) within the actual paper.

2. Can the authors just clarify which multivariate logistic regression was used, it is mentioned it is stepwise, but was it forwards or backwards entered?

3. I may have missed it in the text, but re: the CT performed preoperatively, I note that the median time was 14 days, but why was it done? i.e. as part of the preoperative staging? Secondly re: CT postoperatively, I note that the exclusion criteria was any CTs beyond 50 days were excluded, but again in your unit did all your patients have CT scans post operatively, or was it due to post operative management looking for a complication?

4. I think the title is slightly misleading, as the title suggests that the CT scans preoperatively can identify potential patients who develop postoperative pancreatic fistula - however even in the abstract the finding of patients with pathologies other Pancreatic adenocarcinoma or pancreatitis is a significant finding (but not a radiological diagnosis!). Hence I would probably alter the title to reflect factors that may lead to POPF instead??

Reviewer #4: The authors assessed the risk factor of postoperative pancreatic fistula in patients who underwent pancreaticoduodenectomy (PD). Multivariate analysis showed the larger area of visceral fat and pathology other than pancreatic ductal adenocarcinoma (PDAC) or pancreatitis as risk factor of clincially-relevant POPF (CR-POPF). Authors reported that anthropometric measurement might be useful in detecting high risk patients with CR-POPF.

The following issues need to be thoroughly reviewed.

Major problems

1. Authors should analysis the previous established risk factor of CR-POPF such as drain amylase fluid on postoperative day 1 if they use final diagnosis of the tumor.

2. Authors should show the clinical characteristics of patients who with visceral obesity and those without.

3. In discussion (Page 11, Line 18), author speculated that the difficulty of operation in patients with greater visceral fat is the reason of high rate of CR-POPF. Do authors have objective data that shows the difficulty of pancreas anastomosis.

4. Moreover, do authors have objective data that excessive visceral fat is associated with insulin resistance and other morbidities in their study.

5. In discussion (Page 12, Line 18), authors stated “The mechanism by…soft pancreas parenchyma”. Is it correct? Many studies reported that PDAC decreased the risk of POPF.

6. Please describe about drain management in Method. Did authors place drain in all patients? And, did authors measure amylase level of drain fluid in all patients?

7. In statistics, authors used P-values of <0.05 on univariate analyses to select factors that were entered into multivariate analysis. This cut off of p-value is not common so that author should explain why they used this method.

7. PLOS authors have the option to publish the peer review history of their article (what does this mean?). If published, this will include your full peer review and any attached files.

Reviewer #3: **Yes: **Franscois Runau

Reviewer #4: No

---

## [Author Response · Author response to Decision Letter 1]

6 Nov 2020

Reviewer #3.

1. I agree with reviewer 1, that points out that pancreatic texture hardness/softness is a subjective measure, though from clinical experience I agree that a 'soft' pancreas is more likely to lead to a leak. I accept that the authors have responded to the reviewer's comments: I would add that perhaps the authors add a sentence or two in the limitations of their work that this is a subjective measure, and more work needs to be done to be able to characterise pancreatic tissue density (and also reference the work relating to MR elastogram etc) within the actual paper.

Thank you for your advice. We added that the subjective nature of the surgeon's perception of the pancreas’ texture was a potential limitation in this study and that further research is warranted to quantitatively characterize the pancreatic texture using methods such as MR elastography or US elastography.

2. Can the authors just clarify which multivariate logistic regression was used, it is mentioned it is stepwise, but was it forwards or backwards entered?

We used the backward stepwise method for multivariable logistic regression analysis. This has been included in the statistical analysis portion of the text.

3. I may have missed it in the text, but re: the CT performed preoperatively, I note that the median time was 14 days, but why was it done? i.e. as part of the preoperative staging? Secondly re: CT postoperatively, I note that the exclusion criteria was any CTs beyond 50 days were excluded, but again in your unit did all your patients have CT scans post operatively, or was it due to post operative management looking for a complication?

The CT scans were performed as a preoperative assessment in all subjects. Regarding the exclusion criteria, the “interval between CT and surgery >50 days” was deemed to be misleading. The interval was corrected to 40 days in the previous revision. Therefore, we worded the sentence as “interval between preoperative CT and surgery >40 days” in the text. All subjects routinely underwent postoperative CT scanning for postoperative complications. 

4. I think the title is slightly misleading, as the title suggests that the CT scans preoperatively can identify potential patients who develop postoperative pancreatic fistula - however even in the abstract the finding of patients with pathologies other Pancreatic adenocarcinoma or pancreatitis is a significant finding (but not a radiological diagnosis!). Hence I would probably alter the title to reflect factors that may lead to POPF instead??

Thank you for the suggestion. Accordingly, we changed the title to "Preoperative CT anthropometric measurements and pancreatic pathology increase risk for postoperative pancreatic fistula in patients following pancreaticoduodenectomy" to reflect factors that could cause POPF.

Reviewer #4

1. Authors should analysis the previous established risk factor of CR-POPF such as drain amylase fluid on postoperative day 1 if they use final diagnosis of the tumor.

According to the 2016 update issued by the International Study Group of Pancreatic Fistula, POPF can be diagnosed when any measurable volume of drain fluid on or after postoperative day (POD) 3 with amylase levels exceeding 3 times the upper limit of normal amylase for each specific institution is detected. Although several studies have been conducted to understand the impact of day 1 drain amylase in predicting POPF after pancreaticoduodenectomy, they all presented different cutoff values for day 1 drain amylase concentration [1]. Molinari et al. [2] reported that a drain amylase value >5000 U/L was a predictive factor for pancreatic fistula development. Jin et al. [3] suggested an amylase level of 2365 U/L in the drainage fluid as the optimal cutoff value for predicting pancreatic fistula.

Drain amylase fluid levels on POD 1 may indeed be a useful marker for the identification of POPF. However, further evaluation is needed to determine the optimal cutoff value. Moreover, since other well-established risk factors for POPF, such as sex, body mass index (BMI), pancreatic duct size, pancreatic texture, blood transfusion, intraoperative blood loss, and operation time, were included in our study, we believe the result to be meaningful.

2. Authors should show the clinical characteristics of patients who with visceral obesity and those without.

Thank you for your advice. We have added a supplementary table containing the clinical characteristics of the subjects with and without visceral obesity.

Please refer to the attached file 'S2 Table (DOCX)'.

3. In discussion (Page 11, Line 18), author speculated that the difficulty of operation in patients with greater visceral fat is the reason of high rate of CR-POPF. Do authors have objective data that shows the difficulty of pancreas anastomosis.

Obesity has negative effects on the surgical outcome following pancreaticoduodenectomy according to other studies. [4–5] The technical difficulty of treating obese patients may contribute to the development of POPF. Our study also showed that the operation time was significantly longer in subjects with visceral obesity than in those without. Future research focusing on the effect of visceral obesity on pancreatic fistula after pancreatic resection is needed.

4. Moreover, do authors have objective data that excessive visceral fat is associated with insulin resistance and other morbidities in their study.

Thank you for your comment. There is no objective data on this topic in our research because our study focused primarily on the risk factors for POPF. However, many scientific review papers have assessed the role of visceral obesity as an emerging risk factor for type 2 diabetes, atherosclerosis, and cardiovascular disease. More research is needed on the mechanism by which visceral obesity causes POPF. However, we have incorporated additional references into the text.

5. In discussion (Page 12, Line 18), authors stated “The mechanism by…soft pancreas parenchyma”. Is it correct? Many studies reported that PDAC decreased the risk of POPF.

Thank you for your observation. This was a mistake, and we have replaced it with “pathologies other than PDAC or chronic pancreatitis.”

6. Please describe about drain management in Method. Did authors place drain in all patients? And, did authors measure amylase level of drain fluid in all patients?

At our institution, external drains were inserted in proximity to the pancreatic anastomosis during each surgery, at which time the amylase level of drain fluid was routinely measured. We have added this information to the manuscript.

7. In statistics, authors used P-values of <0.05 on univariate analyses to select factors that were entered into multivariate analysis. This cut off of p-value is not common so that author should explain why they used this method.

Multivariate analysis using all variables are ideal because the statistical program automatically applies and analyzes the significant variables, but its suitability depends on the situation. When many explanatory variables are present, only variables with a p-value of 0.05 or less (i.e., statistically significant variables) as determined by univariate analysis could be included in the multivariate analysis. Researchers can perform statistical analysis in several different ways; however, as pointed out by the reviewer, we re-analyzed the results. When multivariate analysis was performed on all variables, the following results were obtained, and visceral fat and pathology were still significant variables.

Please refer to the table attached in the 'Response to Reviewers (DOCX)' file.

References

1. Liu Y, Li Y, Wang L, Peng CJ. Predictive value of drain pancreatic amylase concentration for postoperative pancreatic fistula on postoperative day 1 after pancreatic resection: An updated meta-analysis. Medicine (Baltimore). 2018;97(38):e12487. https://doi.org/10.1097/MD.0000000000012487 PMID: 30235751

2. Molinari E, Bassi C, Salvia R, Butturini G, Crippa S, Talamini G, et al. Amylase value in drains after pancreatic resection as predictive factor of postoperative pancreatic fistula - Results of a prospective study in 137 patients. Annals of Surgery. 2007;246(2):281-287. https://doi.org/10.1097/SLA.0b013e3180caa42f PMID: 17667507

3. Jin S, Shi XJ, Wang SY, Zhang P, Lv GY, Du XH, et al. Drainage fluid and serum amylase levels accurately predict development of postoperative pancreatic fistula. World J Gastroenterol. 2017;23(34):6357-6364. https://doi.org/10.3748/wjg.v23.i34.6357 PMID: 28974903

4. Park CM, Park JS, Cho ES, Kim JK, Yu JS, Yoon DS. The effect of visceral fat mass on pancreatic fistula after pancreaticoduodenectomy. J Invest Surg. 2012;25(3):169-173. Https://doi.org/10.3109/08941939.2011.616255 PMID: 22583013

5. Shamali A, Shelat V, Jaber B, Wardak A, Ahmed M, Fontana M, et al. Impact of obesity on short and long term results following a pancreatico-duodenectomy. Int J Surg. 2017;42:191-196. https://doi.org/10.1016/j.ijsu.2017.04.058 PMID: 28461146

---

## [Editor Report · Decision Letter 2]

23 Nov 2020

Preoperative CT anthropometric measurements and pancreatic pathology increase risk for postoperative pancreatic fistula in patients following pancreaticoduodenectomy

PONE-D-20-24280R2

Dear Dr. Kim,

We’re pleased to inform you that your manuscript has been judged scientifically suitable for publication and will be formally accepted for publication once it meets all outstanding technical requirements.

Kind regards,

Ulrich Wellner, PD Dr. med.

Academic Editor

PLOS ONE
---

## [Editor Report · Acceptance letter]

25 Nov 2020

PONE-D-20-24280R2 

Preoperative CT anthropometric measurements and pancreatic pathology increase risk for postoperative pancreatic fistula in patients following pancreaticoduodenectomy 

Dear Dr. Kim:

I'm pleased to inform you that your manuscript has been deemed suitable for publication in PLOS ONE. Congratulations! Your manuscript is now with our production department. 

Kind regards, 

on behalf of

Dr. Ulrich Wellner 

Academic Editor

PLOS ONE